# Human Virome and Disease: High-Throughput Sequencing for Virus Discovery, Identification of Phage-Bacteria Dysbiosis and Development of Therapeutic Approaches with Emphasis on the Human Gut

**DOI:** 10.3390/v11070656

**Published:** 2019-07-18

**Authors:** Tasha M. Santiago-Rodriguez, Emily B. Hollister

**Affiliations:** Diversigen Inc., 2450 Holcombe Blvd, Suite BCMA, 77021 Houston, TX, USA

**Keywords:** microbiome, phage therapy, viral mock communities, virome

## Abstract

The virome is comprised of endogenous retroviruses, eukaryotic viruses, and bacteriophages and is increasingly being recognized as an essential part of the human microbiome. The human virome is associated with Type-1 diabetes (T1D), Type-2 diabetes (T2D), Inflammatory Bowel Disease (IBD), Human Immunodeficiency Virus (HIV) infection, and cancer. Increasing evidence also supports trans-kingdom interactions of viruses with bacteria, small eukaryotes and host in disease progression. The present review focuses on virus ecology and biology and how this translates mostly to human gut virome research. Current challenges in the field and how the development of bioinformatic tools and controls are aiding to overcome some of these challenges are also discussed. Finally, the present review also focuses on how human gut virome research could result in translational and clinical studies that may facilitate the development of therapeutic approaches.

## 1. What is the Microbiome?

The human microbiome is comprised of communities of commensal, symbiotic and pathogenic bacteria, viruses, archaea, and small eukaryotes that actively interact with each other and the host to maintain homeostasis [1]. The microbiome is acquired at birth, is shaped by mode of delivery, and consequently diversifies as a result of multiple factors, including diet (e.g., breast feeding and solid food) and other environmental exposures [2]. The human gut microbiome harbors genes that are involved, and in many cases are essential, in nutrient synthesis, as well as in the metabolism of amino acids [3], carbohydrates [4], and lipids [5]. Factors such as host genetics [6], dietary habits [7], lifestyle [8], comorbidities [9], chemotherapy [10], and antibiotics [11], can disrupt this balance and may promote disease. The microbiome field is actively working to understand microbial membership, structure, function, and trans-kingdom interactions, as well as cause and effect relationships in the context of health and disease.

Most microbiome studies, to date, have focused on the bacterial component of the human microbiome mainly because the techniques used for the isolation and characterization of bacterial communities are relatively well-developed and standardized compared to those of the archaeal, eukaryal and viral communities [12]. Additionally, the majority of published human microbiome studies have focused on the characterization of bacterial communities utilizing one or several regions of the 16S rRNA gene [13]. Since the 16S rRNA gene is phylogenetically conserved, analysis pipelines for the study of the 16S rRNA gene are widely available and utilized [14,15], and 16S rRNA gene sequence databases are regularly updated and curated. Similarly, archaeal and eukaryal communities can be studied through the sequencing of one or several regions of the 16S rRNA or 18S rRNA genes, respectively [16,17]. By using shotgun metagenomic sequencing, microbial community membership can also be studied through the sequencing of all genes, rather than the 16S or 18S rRNA genes alone. This approach also possesses the advantage of understanding gene function in association with health, disease, and other types of dysbiosis [18].

Characterizing viral communities can be more challenging than bacteria, archaea and eukaryotes given that viruses do not possess phylogenetically conserved genes. Therefore, shotgun metagenomic sequencing is the preferred approach to characterize viral communities, a concept known as the virome. While viruses are the most numerous (about 10^31^ viral particles on Earth and approximately 10^8^ to 10^9^ per gram of feces) and diverse microbial entities, relatively few studies have focused on the membership and function of viruses as part of the human microbiome [19]. This is probably due to the challenges encountered in viral isolation, nucleic acid extraction, sequencing and analysis pipelines [20]. In the present review, we mainly focus on the role of the gut virome in health, its newly identified potential role(s) in various diseases, some of the challenges encountered in virome characterization, and how consideration of the virome is enabling translational and clinical discoveries. Most of the evidence presented here is focused on the human gut virome when available; nevertheless, information can be translated to the role of viruses and the virome of various body sites including, but not limited to, the oral cavity, skin, urogenital tract, blood, respiratory tract and liver. While research is shedding light into the role of viruses and the virome in diseases associated with the mentioned body sites and sample types, we will not be discussing them in the present review.

## 2. Viruses as Part of the Human Microbiome

Viruses are obligate parasites with single- or double-stranded DNA or RNA genomes [21]. Viral infection begins when surface proteins bind to receptor proteins on the host cell surface, followed by replication and lysis. Alternatively, some viruses can remain dormant until conditions are favorable for replication and host cell lysis [22]. While protein-protein interactions between virion and host are very specific in many cases, some viruses can have a broader host range. Curation of the human virome is largely based on double-stranded DNA viruses, although a smaller number of studies have focused on the RNA viral fraction, which comprises the majority of the eukaryotic virome [21]. The human virome can include endogenous retroviruses and eukaryotic viruses, which can infect human, plant, or other animal/small eukaryote cells (i.e., dietary components), as well as bacterial viruses (i.e., bacteriophages).

### 2.1. Human Endogenous Retroviruses

Human endogenous retroviruses (HERVs) are “fossil viruses” that account for approximately 8% of the human genome and consist of proviral DNA or partial, integrated genomes [23]. Their detection is usually not through the application of high-throughput sequencing, but rather through sequencing of heterochromatin regions or host DNA [24]. Since HERVs are integrated into the human genome, proviral DNA in germline cells can be transmitted to the offspring [25]. Infectious viral particles are not produced as a result of HERV transcription, but rather, HERV transcription can result in the production of functional proteins [25]. The potential role of HERV proteins in promoting disease is of increasing interest. For instance, diseases, including autoimmune disease (e.g., Multiple Sclerosis (MS), type I diabetes mellitus, and rheumatoid arthritis), neurodegenerative (e.g., Amyotrophic lateral sclerosis (ALS)) disorders, chronic inflammation, and cancer may promote the replication of HERVs [26]. In addition, infection by Herpes virus can also activate HERVs, which has also been associated with MS [26]. The association of HERVs with several disease phenotypes has prompted further research to understand underlying mechanisms through which HERVs may promote disease. For instance, in neurodegenerative disorders such as ALS, HERVs can be expressed in neurons, which can potentially promote disease progression [27]. In autoimmune diseases, it is hypothesized that HERVs affect the expression of regulatory genes, which may also activate the immune system [28]. Relatedly, in cancers such as melanoma, breast cancer, and ovarian cancer, HERVs express proteins that can elicit an immune response [29]. HERV research in the context of disease has also prompted interest in the development of therapeutic approaches which target the transcription of HERV genes. Specifically, it has been proposed that a combination of histone deacetylase inhibitor (HDACi) and checkpoint inhibitors may be a potential approach [29].

### 2.2. Eukaryotic Viruses

In addition to the human endogenous retroviruses that are an intrinsic part of the human genome, the human virome also includes eukaryotic viruses which can infect human cells, smaller eukaryotes (i.e., protozoans and fungi), and the plants and animals that may be ingested as a part of the diet [30]. The human eukaryotic virome is often associated with disease and its global characterization is now facilitated by the application of high-throughput sequencing rather than culture- and PCR-based methods for the detection of specific strains [31]. High-throughput sequencing offers the advantage of eukaryotic virus discovery and comparison with other strains without a priori knowledge. Many human eukaryotic viruses, including those from the *Phycodnaviridae*, *Herpesviridae*, *Poxviridae*, *Mimiviridae* and *Iridoviridae* families, have been identified in babies stool samples, although in much lower relative abundances compared to bacteriophages [32]. Although some of these viruses have not previously been identified as part of the human virome, it is possible that the sequences were misclassified during the annotation process. The same eukaryotic virus strains can be found in twins, and the same viral families can be found in a mother’s breast milk and her baby’s stool, suggesting a direct transmission through the mother-baby bond [33,34]. Certain human eukaryotic viruses can also be transmitted through contaminated food, water, and direct contact with fomites [35,36]. A number of these infections can be common while others are rare. In addition, although some individuals develop symptoms, others can remain asymptomatic and shed viruses for prolong periods of time [37]. High-throughput sequencing has enabled the identification of current and novel strains of eukaryotic viruses. For instance, novel strains of astroviruses were discovered in the stool of children in Australia and children with diarrhea in Virginia, Nigeria, Pakistan, and Nepal [38]. Approaches like this can aid in the development of molecular tools for the rapid detection of novel strains in environmental and human samples to prevent further spread, understand evolution, and facilitate diagnosis through the development of new assays.

High-throughput sequencing can enable the identification of viruses in diseases associated with an expected etiology (Figure 1). For instance, the identification of Bas-Congo virus in subjects with acute hemorrhagic fever using high-throughput sequencing is important because it is the first time that viruses from the *Rhabdoviridae* family have been implicated with the disease [39]. Acute hemorrhagic fever is usually associated with dengue virus, Ebola-Marburg viral diseases, Lassa fever, yellow fever, Rift Valley fever, hantavirus infections, and Crimean-Congo hemorrhagic fever, just to mention a few examples [40]. Similarly, high-throughput sequencing can enable the discovery of viruses associated with diseases of unknown etiology. Such is the case with Kawasaki Disease (KD), a condition which is known to have a significant genetic component for its progression. Although the KD alleles are equally present in subjects of East Asian and Caucasian ancestry, KD occurs more frequently in East Asian populations, suggesting that genetic predisposition alone is insufficient to explain disease. Interestingly, a viral etiology has been implicated with KD. Specifically, with the use of high-throughput sequencing, a variant of torque teno virus 7 (TTV7) was found in subjects with KD, but was absent from control subjects [41] (Figure 1). Although additional studies are needed to confirm the presence of TTV7 in a larger cohort, these findings are intriguing as they suggest a role for uncharacterized viruses in the progression of diseases of unknown etiology. Other examples include novel or unexpected viruses, such as novel human astroviruses causing encephalitis, meningoencephalitis, and meningitis [42]. Broader characterization of the human virome may aid in monitoring potential future outbreaks and understand underlying mechanisms of infection.

With advances in microbiome research, we appreciate that human enteric viruses can benefit from the bacterial members of the microbiome. For instance, human enteric viruses can utilize bacterial surface polysaccharides, which enhance infectivity and pathogenesis [43]. This results in an initiation of immune responses that result in host tolerance, viral replication and finally, transmission of the virus. In mouse models, for instance, gut bacteria promote the replication and transmission of enteric viruses from the *Retroviridae*, *Picornaviridae*, *Reoviridae*, and *Caliciviridae* families [44]. This is intriguing as it provides evidence in support of the influence of trans-kingdom interactions on health and disease.

### 2.3. Bacteriophages

Bacteriophages or phages are the most abundant viral entities and are known to inhabit every niche as their bacterial hosts, including the human gut, either in the form of a prophage, prophage remnant or virion [45]. The gut virome is acquired from an early age and it is known to share some viral components found in maternal milk [32,46], indicating that gut viromes may reflect dietary habits. Although such studies are still very limited, it is known that adult individuals on the same diet have more similar gut virome composition than individuals on different diets [47]. In further support of diet as a potential driver of gut virome composition, fermented foods and dairy products often contain specific viruses, including phages that infect *Leuconostoc*, *Weisella*, *Listonella*, *Escherichia* and *Staphylococcus*, which may affect the endogenous virome [48].

Like their bacterial counterparts, phages often exhibit biogeographic specificity across human body surfaces, including the human gut, skin and oral cavity [49,50]; yet, unlike the bacterial fraction of the microbiome, phages seem to be more subject-specific [51]. This biogeographic specificity can affect the ability of phages to infect specific bacterial species and strains, many of which are also known to possess biogeographic specificity [50]; yet, phages can also be of cosmopolitan nature and exhibit a wide host range [52]. Phages from the human gut belong to the *Caudovirales* order and can be classified into one of the main families including the *Siphoviridae* (phages with long, non-contractile tails, which are largely temperate but does include some lytic members), *Myoviridae* (phages with long, contractile tails, which are strictly lytic)*, Podoviridae* (phages with short, contractile tails, which are largely lytic, but does include some temperate members), and more recently, to the *Ackermannviridae* and *Herelleviridae* families, which are not classified according to morphology, but based on genome structure [53]. Phages can be strictly lytic, where they lack the genes necessary to encode integrases, recombinases, and other genes required for integration/excision of the phage genome from the bacterial host chromosome. Temperate bacteriophages, on the other hand, do not lyse their host immediately, rather, remain dormant in their bacterial host genomes as a prophage until conditions promote the expression of excisionases and related genes [54]. Phage families and lytic/lysogenic cycles may be particularly important when interpreting a microbiome-related dysbiosis. For instance, in periodontal health and disease, a higher prevalence of phages from the *Myoviridae* family has been noted in subgingival plaque of subjects with the condition [55]. A similar outcome has been noted for phages associated with ulcerative colitis (UC), where microviruses, myoviruses and podoviruses are more abundant in subjects with the disease [56]; however, little is known if strictly lytic or temperate phages have a greater potential role in disease progression. Although most phages in various human sites are classified as temperate, this may be due to temperate phages being more represented in databases.

## 3. Lytic and Lysogenic Cycles in the Control of Bacterial Populations

An increasing number of studies have demonstrated that phages have the potential to promote health and disease, through both direct and indirect means. The role of phages in health and disease may intrinsically be associated with their lytic/lysogenic cycles. Pioneering studies of lytic and lysogenic cycles focused on marine phages have demonstrated the ability of certain phage to decrease bacterial numbers in response to temperature [57], and UV radiation [58]. UV radiation, in particular, may have a direct effect on phages, acting as an inducing agent and promoting the expression of genes associated with the lytic cycle. UV radiation can also affect phage inactivation rates, where the greater the UV radiation, the greater the inactivation rate of bacteriophages and vice versa [59].

In the human body, the degree to which pH, oxygen levels, and nutrient availability affect phage membership and function has not been extensively studied, but the identity and relative abundances of enteric phages are known to change in association with health status. The relative proportions of strictly lytic and temperate phages, as well as the genes associated with lytic and lysogenic cycles tend to differ in association with several disease phenotypes; yet, the underlying reasons remain to be investigated individually. It is reasonable to speculate that external factors are associated with the induction of the lytic cycle that may result in the lysis of beneficial bacteria and this may in turn lead to dysbiosis and disease (Figure 2A).

Current knowledge on the role of the phage on some of these conditions will be discussed in this review. Generally, temperate phages will maintain a prophage state when bacterial numbers fall under a specific threshold [60], which may be in agreement with the fact that parasites will not eliminate their hosts completely. In addition, there is a constant arms-race between bacteria and their phage that include restriction-modification systems, CRISPR/Cas and abortive infection [61,62]. A number of reviews have focused on the ongoing evolution between phage and bacteria and will not be discussed in the present review. Temperate phages in natural environments tend to be more abundant under oligotrophic or nutrient limiting conditions. Maintaining a lysogenic state can, in turn, protect bacteria from a superinfection from strictly lytic phages [63].

How this information translates to the human gut and other human surfaces remains a matter of further research and speculation. The presence of a bacterial host cell and a phage counterpart is not sufficient to lead to an infection because the cell must be in an appropriate physiological state and in an accessible anatomical location for an infection to occur. For instance, although stool is utilized as a proxy of the human gut because of the relative ease of sampling, the human gut is a multi-niche ecosystem with differing conditions [60]. In the small intestine, for example, high oxygen levels can select for facultative anaerobes including *Escherichia coli*, *Streptococcus* spp. and *Bacteroidetes* [64]. Stool samples are usually considered to be a reflection of the large intestine microbiome, which is typically composed of facultative or obligate anaerobes belonging to the Firmicutes and Bacteroidetes phyla, *Bacteroidaceae*, *Ruminococcaceae*, *Lachnospiraceae* and *Coriobacteriaceae* families, *Desulfovibrio* spp. and lactic acid bacteria, such as members of the *Leuconostoc* and *Lactococcus* genera [65]. The human gut lumen is also characterized by a mucus layer that supports the growth of a low number of microbes including mucin-degrading bacteria such as *Bacteroides fragilis*. The ecology of bacteria in the human gut can also be affected by the timing and order of succession. For instance, *E. coli* could better persist if they are the first colonizers of adhesion sites [65].

Given the niche-specificity of bacteria across body sites, with the gut being one of the most complex ecosystems, it is reasonable to speculate that phages also exhibit niche-specificity. For instance, very few studies have determined the membership of phages across the different niches of the human gut as most studies have been performed with stool samples [20]. Unlike marine environments, lysogeny is probably the preferred state in stool samples (and possibly across the human gut and other body sites). The rationale lies on the “piggyback the winner” hypothesis, where phages will benefit on maintaining a prophage state as their bacterial hosts replicates, ensuring their own replication and also protection from the environment (i.e., from proteases) [66]. The reason is that various body sites can be dynamic environments with complex differences in anatomy and physiology, and be affected by the constant influx of new microbes, as well as by immunological changes. Interestingly, a number of bacterial populations remain inaccessible to phages in certain body sites, such as the lumen; yet, in mucosal surfaces, phages may act as a type of immune system and maintain a symbiotic relationship, where they will lyse invading bacteria, some of which can be opportunistic pathogens (Figure 2B) [67]. While these studies have utilized in vitro models, results are elucidating phages-bacteria dynamics, and their relationship with the human immune system.

## 4. Bacterial Adaptation through Lysogenic Conversion

Lysogeny provides additional evolutionary advantages to the bacterial host [68]. The conferred benefits are usually accompanied by changes in genome structure and gene expression which enhance adaptation and survival [68]. Virulence factors, antibiotic-resistance and metabolic genes are among the genes that confer evolutionary advantages to the bacterial host. There are a number of specific examples of phage-encoded genes that enable bacterial colonization and adhesion [69,70,71,72]; promote cell invasion [73,74,75]; enable resistance to serum and phagocytes [76,77,78,79,80,81]; are involved in exotoxin production [82,83,84,85,86,87,88,89]; and confer antibiotic susceptibility [90,91] (Table 1; Modified from [92]). Similar to virulence factors promoting the mentioned functions, phages can also carry antibiotic-resistance genes. Interestingly, several antibiotic-resistance genes identified in phage genomes include those encoding for beta-lactamases, penicillin-binding proteins and fluoroquinolone-resistance [93]. While virulence factors and antibiotic-resistance genes may provide benefit to the bacterial host, they may also provide advantage to the phage. For instance, Shiga toxin 2 (Stx2)-encoding phages (as well as their bacterial hosts) can better survive in natural environments [94], better resist chlorination [95] and UV [59], and utilize *stx2* as a defense mechanism against predation from *Tetrahymena thermophila* [96], suggesting that *stx2* may be intrinsically involved in survival. In addition, certain sequences encoding for virulence factors are part of the phage structural genes. This is the case of the Ace protein, which is responsible, in part, for *Vibrio cholerae* enterotoxicity and constitutes a phage minor structural protein. These data suggest that a number of virulence factors and antibiotic-resistance genes encoded by phages may incidentally enable additional fitness features to the bacterial host [92].

Knowledge regarding the role of phages in transferring virulence factors, antibiotic-resistance genes and other beneficial genes to individual bacterial species under in vitro conditions may aid in the understanding of the role(s) of lysogenic conversion in the human gut (Figure 3). Given that the human gut is a diverse and competitive ecosystem with multiple niches, varying oxygen levels, nutrient resources and surface area available for colonization. Thus, it is reasonable to hypothesize that lysogenic conversion in the human gut and other human surfaces enables the survival of both bacterial hosts and phages under differing and stressful conditions.

## 5. Bacteriophages as Human Pathogens?

In addition to controlling bacterial populations and promoting an evolutionary advantage to the bacterial host through lysogenic conversion, phages have also been proposed as human pathogens. This was probably first suggested with phages harboring stx2 [97]. Stx poses a serious and, in many cases, lethal risk to human health [98]. Human endothelial cells possess a receptor known as globotriaosylceramide (Gb3) which is targeted by the B subunit of Stx [97]. It remains uncertain how the toxin enters systemic circulation, but evidence from in vitro studies suggests that prophages harboring *stx* may be induced and released into the human gut. Stx-encoding phages could then be translocated across the epithelial barrier in or on blood cells including erythrocytes, platelets, granulocytes, monocytes and lymphocytes. Although possible, this hypothesized mechanism remains largely controversial as in vitro studies have not been able to fully demonstrate this with Stx-encoding phages [97]; however, recent data have demonstrated translocation through transcytosis of T4 phages under in vitro conditions [99]. This suggests that, although phages do not recognize eukaryotic cells directly, there are mechanisms that may be directly involved in their movement across the human gut barrier.

Intestinal permeability may be another mechanism of phage movement in the human gut associated with different disease phenotypes. Phages can indirectly increase gut permeability by directly affecting the bacterial communities involved in maintaining intestinal integrity and thus, may in turn indirectly act as human pathogens. This has been demonstrated in a murine model, and it is expected that a similar process could occur in humans [100]. Following the introduction of phages infecting bacteria from the *Enterobacteriaceae*, *Staphylococcaceae* and *Streptococcaceae* families to mice, an increase in lactulose/mannitol ratio and in *Butyrivibrio*, *Oscillospira* and *Ruminococcus* relative abundances, as well as a decrease in the relative abundances of *Blautia*, *Catenibacterium*, *Lactobacillus* and *Faecalibacterium* was observed. The decrease in *Lactobacillus* and *Faecalibacterium*, specifically, is associated with altered short chain fatty acid production, impaired gut permeability, and inflammation (Figure 4) [100]. Although this could result in the transport of prophage through transcytosis, as described above, movement of phage may also take place as a function of direct passage through an impaired intestinal barrier. Through interaction with gut bacterial community members and the host itself, phages can alter intestinal permeability and therefore, promote disease.

## 6. Virome-Associated Dysbiosis

Dysbiosis at sites across the human body, including the gut, is associated with disease. Although translational animal models are used to understand cause-effect relationships, virome analyses of the human gut are providing evidence on the potential role(s) of viruses in maintaining homeostasis or promoting disease. We will discuss current knowledge regarding the role of the virome in association with Type-1 diabetes (T1D), Type-2 diabetes (T2D), Inflammatory Bowel Disease (IBD), HIV infection and cancer.

### 6.1. Type-1 Diabetes

T1D is a proinflammatory disease involving an autoimmune attack on the insulin-secreting beta cells of the pancreatic islets of Langerhans, resulting in a loss of these cells [101]. The disease can be identified by the presence of antibodies against beta-cell autoantigens including insulin, zinc transporter 8 and islet antigen 2. T1D possesses a genetic component, with over 50 genes being associated with the condition [101]. Changes in lifestyle, particularly a westernized diet (i.e., high-fat, high-carbohydrate and low-fiber diet) have also been associated with T1D progression [102]. Dietary habits are known to alter the gut microbiome, and some studies suggest that altered gut microbiome composition may be associated with T1D progression. Specifically, *Prevotella*, *Faecalibacterium*, *Eubacterium*, *Fusobacterium, Anaerostipes*, *Roseburia* and *Subdoligranulum* have been described as being more abundant in healthy controls compared to T1D subjects [103]. In contrast, *Lactobacillus*, *Lactococcus*, *Bifidobacterium* and *Streptococcus* were more abundant in T1D subjects.

Murine models of T1D have shown that a virus infection can damage the pancreatic islets of Langerhans [104], suggesting the potential for a viral association with T1D. In a study involving children with T1D, viruses from the *Circoviridae* family were more prevalent in the control group compared to children with T1D; yet, not all the children in the control group showed the presence of circoviruses. The study did not identify other significant differences with respect to the relative abundances of eukaryotic viruses, but the phageome has also been investigated in subjects with T1D. No significant differences were noted in the relative abundances at the family level in the *Microviridae*, *Myoviridae*, *Podoviridae* and *Siphoviridae* families; yet, *Myoviridae* evenness (Shannon diversity) and richness were decreased in subjects with T1D and tended to be increased in subjects with no T1D. A similar trend was noted with podoviruses, which were more represented in children without T1D [105]. Current studies show that relationships between specific phages and their corresponding bacterial host in association with T1D are complex and may require additional in vitro analyses. Recently, it has been suggested that amyloid-producing *E. coli* and their phages may increase the risk of children to develop T1D. In children who presented seroconversion or developed T1D, an increase in the *E. coli* phage/*E. coli* ratio prior to bacterial host depletion was observed, suggesting that this may be due to prophage induction [106].

### 6.2. Type-2 Diabetes

T2D is an increasing concern to public health, particularly in Westernized societies, and is characterized by hyperglycemia associated with insulin-resistance [107]. T2D progression is intimately associated with both genetic and environment factors, particularly dietary habits, and most recently has been associated with altered microbiome composition. Significant reductions in Firmicutes and the order *Clostridia* have been observed in subjects with T2D [107]. Follow-up studies identified several *Clostridium* spp., as well as *Akkermansia muciniphila*, *Bacteroides intestinalis* and *E. coli* being enriched in subjects with T2D [108]. These data suggested an association between specific members of the human gut microbiome and T2D. Importantly, an increase in the median relative abundance of phages predicted to infect *Escherichia* and *Clostridium* belonging to the *Podoviridae* family (strictly lytic and some are temperate), as well as those infecting *Lactobacilllus*, *Pseudomonas* and *Staphylococcus* belonging to the *Siphoviridae* family (temperate and some are strictly lytic) in subjects with T2D was observed [109]. Although the study did not specifically isolate viruses prior sequencing, results are still intriguing. The benefit of analyzing sequence data from shotgun metagenomic datasets without a prior isolation of viruses is that it could enable the study of viral sequences that are still integrated in the bacterial host genomes [109]. An enrichment of phages infecting *Escherichia* and *Clostridium*, and an enrichment of phages infecting the corresponding genera could suggest that: (i) the identified phages may infect *Escherichia* and *Clostridium* spp. other than those enriched in subjects with T2D, and that (ii) the identified phages, although belonging to be *Podoviridae* family, may be temperate phages that were propagated along with their bacterial hosts in association with T2D. These data suggest the presence of complex trans-kingdom interactions between bacteria and their viruses (lytic and temperate) in association with T2D, which could also be used as potential markers of the disease and/or therapeutic targets.

### 6.3. Inflammatory Bowel Disease (IBD)

IBD is an immune-mediated disease that causes inflammation of the gastrointestinal tract and includes Crohn’s disease (CD) and ulcerative colitis (UC), which can require medication and in extreme cases, surgery. Several factors including genetic, environmental, immune, dietary and microbial are associated with IBD. Intestinal inflammation associated with IBD has been implicated with a decrease in beneficial bacteria. Bacterial signatures can differ based on IBD type (CD vs UC), and whether surgery has been performed [110]. Broadly, these bacterial signatures include a reduction in *Faecalibacterium prausnitzii*, *Roseburia* spp., *Bifidobacterium* spp. and *Lactobacillus* spp., as well as an expansion of *E. coli*, *Clostridium difficile*, *Oscillospira* and unclassified *Ruminococcaceae* [111,112]. Microbial signatures of IBD are essential because they may represent biomarkers of the disease, or even potential therapeutic targets. In addition, in vivo studies using animal models are needed to understand cause-effect relationships in IBD.

Several studies have investigated the association of the enteric virome with IBD. Although studies thus far have produced somewhat conflicting results, they have revealed alterations of the enteric virome in association with IBD that merit further investigation. One of the first studies to assess the potential association between the enteric virome and IBD showed an increased abundance of phages infecting *Clostridiales*, *Alteromonadales* and *Clostridium acetobutylicum*, as well as viruses from the *Retroviridae* family being more abundant in subjects with IBD [113]. A follow-up study which included household controls found differing results [114]. Specifically, a significant higher viral diversity from phages was found in subjects with IBD compared to household controls, and this increased viral diversity was accompanied by a decrease in bacterial diversity. A similar outcome was observed in a pediatric population, where viruses from the *Caudovirales* order were more represented in subjects with IBD [115]. Another study characterizing the virome of the gut mucosa of a murine model of colitis showed an increase in the abundance of viruses from the *Caudovirales* order, but a decrease in their diversity [116]. Interestingly, phages infecting enterobacteria were significantly more represented in mice with colitis. The higher abundance and decreased diversity of phages is in agreement with a reduced number of phage-related functions related with UC [116]. These results altogether are opening the possibility of exploring therapeutics to target the virome in IBD subtypes.

### 6.4. Human Immunodeficiency Virus (HIV) Infection

HIV infection is estimated to affect over 35 million people worldwide and can result in acquired immunodeficiency syndrome (AIDS) and the further development of opportunistic infections and cancer [117,118]. HIV infection can also result in a number of comorbidities including inflammation and diarrhea, and studies suggest that gut microbiome dysbiosis is associated with HIV infection [119]. Certain bacterial families seem to discriminate between HIV-positive individuals with CD4 T-cell counts < 200 versus CD4 T-cell counts > 200. Some of these bacterial families include *Enterobacteriaceae, Pasteurellaceae, Streptococcaceae, Enterococcaceae, Lactobacillaceae, Planococcaceae, Actinomycetalaceae, Carnobacteriaceae, Micrococcaceae, Gemellaceae, Comamonadaceae, Leuconostocaceae* and *Leptotrichiaceae*. Although these results are not indicative of cause/effect of HIV-associated comorbidities, they may provide insight into treating and/or ameliorating HIV-associated comorbidities [119].

The gut virome is also affected by HIV infection. Particularly, adenoviruses were significantly more represented in HIV-positive subjects with CD4 T-cell counts < 200 when compared to HIV-negative subjects and those with CD4 T-cell counts > 200. This same study showed an increase in the *Anelloviridae* family in association with HIV infection, which may be driven by CD4 T-cell counts < 200. Surprisingly, although certain bacterial communities were affected by CD4 T-cell counts, phage richness and evenness were not affected [119]. This study demonstrates that the phageome may not always be affected by changes in bacterial composition. Another study found a similar outcome, where the relative abundance of viruses from the *Anelloviridae* family was negatively correlated with CD4 T-cell counts [120]. These data were also limited to the DNA virome, suggesting that the RNA fraction of the virome needs to also be considered in microbiome-disease studies.

### 6.5. Cancer

Pioneering studies investigating the association of specific microorganisms and cancer have led to tremendous implications for public health. For instance, *Helicobacter pylori* and its association with gastric cancer aided in the classification of this bacterium as a class I carcinogen. Viruses associated with different types of cancer include Epstein-Barr virus (EBV), Hepatitis B, Hepatitis C, HIV-1, Human Papillomavirus (HPV) and Human T-cell lymphotropic virus type I (HTLV-1), which are also classified as class I carcinogens [121]. Some of these viruses may not be considered part of the human virome due to their implications with cancer; however, certain viruses, including HPV, may be considered part of the human virome as those considered to be low- and high-risk, and those with unknown pathogenicity have been identified in asymptomatic populations. Moreover, only a small percentage of those subjects infected with high-risk HPV develops cancer [122]. Other ubiquitous viruses that can cause cancer include Human polyomavirus [123]. Knowledge regarding eukaryotic viruses and disease tends to be more advanced than our understanding of the relationships between disease and gut-microbiome. This knowledge has also provided key information regarding the mechanisms through which viral infection can cause various cancer types, including immunodeficiency, chronic inflammation, and virus oncogenes. In addition, an increasing number of studies are investigating the associations between the gut microbiome and specific bacterial members with different cancers including melanoma [124,125,126], non-Hodgkin’s lymphoma [10], cervical cancer [127], Acute Lymphoblastic Leukemia (ALL) [128], and colorectal cancer (CRC).

The association of CRC and the microbiome may be one of the most studied compare to other cancer types. CRC is frequently characterized by decreased bacterial diversity in feces and mucosal samples [129]. This decreased diversity is often accompanied by the absence of certain bacteria that may be implicated in maintaining a healthy state and the detection of taxa associated with CRC and tumorigenesis, including *Fusobacterim nucleatum*, *Desulfovibrio* spp., *Bilophila wadsworthia*, *Parvimona, Alistipes* and *E. coli* [130,131,132,133]. These bacteria are being evaluated as potential candidates for therapeutic approaches [124,125,126] and potential diagnostics. In addition, gut microbiome profiles may be used in the future to predict if a patient may respond to treatment.

Although bacteria are known to have roles in cancer progression, the association and potential causation of viruses such as EBV, Hepatitis B, Hepatitis C, HIV-1, HPV and HTLV-1 with different types of cancer has prompted the investigation of similar role(s) of the enteric virome. Although the eukaryotic viral faction of the enteric virome has been associated with CRC to a lesser extent, phage communities have the potential to be the most influential in this context. Phages may not necessarily lyse the most influential bacteria associated with CRC, rather certain phages appear to act as “community hubs”, influencing CRC by altering overall bacterial community composition [134]. Similar to specific bacterial signatures, viral signatures have been identified in CRC, mostly being bacteriophages from the *Siphoviridae* and *Myoviridae* families [134]. Another study that did not isolate viruses specifically, found >20 viral genera that discriminated between subjects with CRC and healthy controls [135]. Eukaryotic viruses that were significantly more represented in subjects with CRC included *Orthobunyavirus*, *Tunavirus*, *Phikzvirus*, *Betabaculovirus* and *Zindervirus.* Other viruses like *Fromanvirus* seemed to be represented only in the healthy cohort. In terms of the phageome, *Streptococcus* phage SpSL1, *Streptococcus* phage 5093, *Streptococcus* phage K13, *Vibrio* phage pYD38-A and Enterobacteria phage HK544 were more abundant in subjects with CRC [135]. Alterations in the gut virome represent the opportunity to elucidate bacterial-viral-host interactions in promoting CRC.

## 7. Trans-Kingdom Interactions

Although we have discussed each viral component of the microbiome in disease individually, it should be noted that each is constantly interacting with bacteria, smaller eukaryotes, and host cells (e.g., mouse and human). These complex trans-kingdom interactions have the potential to govern health and disease at various human body sites [44]. It is anticipated that trans-kingdom interactions may underly disease mechanisms in cases where the biology cannot be explained by a single component of the microbiome. Thus far, several examples of this phenomenon include interactions between eukaryotic viruses with bacteria, helminths, and host, as well as bacteriophages with bacterial host and host cells.

These interactions are essential for virus transmission, replication and evasion of the immune system. For instance, mouse mammary tumor virus (MMTV) are transmitted more efficiently in mice with undisturbed microbiomes, but antibiotic-treated mice were uncapable of transmitting MMTV to their offspring [136]. Results from this study also suggest that MMTV may evade the host immune system under certain circumstances, as newborn mice that ingested the virus did not produce antibodies [136]. MMTV can also interact with bacteria. Specifically, MMTV possesses a lipid bilayer envelope with lipopolysaccharides (LPS)-binding factors CD14, TLR4 and MD-2 that binds to LPS on the surface of *Bacteroides* [137]. This has also been shown for polioviruses, where they can bind to LPS, prevent premature RNA release and enhance viral binding [138]. Helminth infection (*Trichinella*) can result in a decreased immune response to murine norovirus (MNoV) infection [139]. Interestingly, MNoV infection has also been shown to increase host susceptibility to diseases such as T1D, MS and IBD. Although there is a genetic pre-disposition to these conditions, the reason some subjects develop disease and others do not remains unclear. Interestingly, both MNoV and a mutation in the *ATG16L1* gene can increase the risk for CD [140]. Finally, as discussed above, bacteriophage-bacteria and bacteriophage-host interactions are also examples of trans-kingdom interactions that may impact health and disease. Recently published results indicate that the presence of the Pf4 phage in *Pseudomonas aeruginosa*-infected skin wounds interferes with host immune signaling by inhibiting tumor necrosis factor production, which prevents phagocytosis and leads to chronic infection [141].

## 8. Challenges in Virome Research

Although the association of the gut virome with various disease states is intriguing, the field is subject to technical limitations. Primary among these is the inability to annotate a large proportion of viral sequence data. It is estimated that 40% to 90% of the viral sequences generated in a given study cannot be annotated, limiting the ability to resolve the identity and functional genes of the viral “dark matter (i.e., viral sequences that do not align to any current viral sequence database) [142]. This may be due to highly divergent and highly evolving viruses, as well as viruses that have yet to be discovered. The challenge of characterizing the viral “dark matter” should be addressed using different approaches including, but not limited to, concentration and purification of virus and virus-like particles using sequential filtration and/or CsCl gradient ultracentrifugation [143], host cell depletion (e.g., animal, plant or human host and bacteria) prior to viral nucleic acid extraction [144], enrichment of viral nucleic acids, removal of host DNA sequences after sequencing [145], and application of an appropriate taxonomic annotation tool, which will be discussed below. A number of reviews have addressed limitations with wet-lab protocols previously [142] and will not be discussed here. Rather, this review will present several of the currently available bioinformatic tools for taxonomic and functional annotation of viral sequences and highlight those that could circumvent some of the current challenges associated with viral taxonomic and functional annotation.

### 8.1. Bioinformatic Methods for Virome Analysis

There is currently no stand-alone pipeline available for virome analysis similar to those developed for 16S rRNA gene sequence analyses. Since most bioinformatic pipelines for virome analysis rely on sequence alignment to current databases, up to 90% of sequences may go unannotated [142]. This limits the ability to understand virome structure and function in a number of different environments, as well as in association with health and disease. Various analysis tools, mostly for taxonomic characterization have become publicly available in the last couple of years. While most rely on direct sequence comparisons to current databases, other tools classify sequences based on k-mers. Table 2 shows a number of these tools based on sequence alignments [146,147,148,149,150,151,152,153,154,155,156,157,158] and k-mer analyses [152,159,160]. Each tool possesses advantages and disadvantages, and each should be evaluated in the context of sample type and goal(s) of one’s study. Some of these tools also require assembled sequences, which adds an additional challenge since differing results have been observed when applying assembly-based methods and different assemblers themselves [161]. The use of controls, particularly mock communities, can aid in addressing some of the biases that could be introduced in virome research to help elucidate the most suitable bioinformatic pipeline.

### 8.2. Viral Mock Communities as Controls in Virome Research

Mock communities have strongly been suggested as controls in microbiome research because they can be used to address biases that may be introduced at different stages of microbiome research, including sample collection and storage, nucleic acid extraction, library preparation, sequencing and data analysis [162]. When outputs describing mock community composition do not reflect that of their starting material, they signal an opportunity to optimize protocol(s) and understand the limitations associated with specific techniques.

Most publicly or commercially available mock communities consist of a mix of bacteria (whole cells or DNA), with a combination of both Gram-negative and Gram-positive in specific proportions [163]. Other criteria for the preparation of mock communities include adding bacteria with high-GC content, low-GC content, small genomes and large genomes [163]. Mock communities mimicking various body sites including the gut, oral, skin and vaginal microbiomes have also been developed and are commercially available through the American Type Culture Collection (ATCC), providing a more customized option of the microbiome of interest. This provides the opportunity to address the issues encountered when studying site-specific microbiomes. Chromosomally-tagged bacteria with unique short sequences (i.e., various regions of the 16S rRNA gene) that can be spiked into the microbiome sample of interest are also available through ATCC and provide an alternative to mock communities of defined composition. A number of researchers and laboratories prepare mock communities for their own internal use [164]. Some of these options can include similar versions of the publicly or commercially available mock communities, as well as non-human mock communities consisting of bacteria (either whole cells or DNA) inhabiting extreme or unique environments (e.g., hypersaline and plant pathogens) that are also to be spiked into human microbiome samples [165]. Bacterial mock communities are also important in virome research as virus characterization is usually accompanied by the characterization of the bacterial fraction of a microbiome sample of interest.

Most viral mock communities described in the literature have been prepared for internal use, and only one viral mock community was found to be currently commercially available during the preparation of this review. This viral mock community, prepared and offered by the ATCC consists of human eukaryotic viruses including human mastadenovirus F, human herpesvirus 5, human respiratory syncytial virus, influenza B virus, reovirus 3, and Zika virus. This viral mock community may be best utilized with clinical samples or studies aiming to study the virome for diagnostic purposes. Similar internally-prepared mock communities have consisted of human mastadenovirus B, human mastadenovirus C, murine gammaherpesvirus 4, coxsackievirus B4 (strain Tuscany), echovirus E13 (strain Del Carmen), human poliovirus type 1 (strain Mahoney), and rotavirus A [150]. Other viral mock communities prepared for internal use have included both eukaryotic viruses and phages [166], such as Vaccinia Western Reserve virus, Lambda phage, Human adenovirus 5, φ29 phage, M13 phage, Minute Virus of Mice p, and a Porcine circovirus 2a. This mock community may also serve as a control during the characterization of the phageome in a microbiome sample of interest. Viral mock communities may also aid in the optimization of protocols for virus recovery including host-cell and host DNA depletion, low nucleic acid yields, amplification biases, sequencing depth and bioinformatic tools utilized for taxonomic and functional annotation.

## 9. Virus- and Virome-Directed Therapeutic Approaches

As virome research expands and becomes more standardized, results are providing the opportunity to utilize virus- and virome-directed therapeutic approaches to treat various diseases [20]. For instance, lytic phages have been used to treat bacterial infections since the early 1900’s, but their use was quickly replaced with the discovery of penicillin [167]. The use and abuse of penicillin and other antibiotics has resulted in the emergence of antibiotic-resistant bacteria. Infections caused by antibiotic-resistant bacteria account for substantial morbidity and mortality worldwide [168]. Phage therapy, which is defined as the use of lytic phages to treat bacterial infections, has been recently re-visited due to an increase in antibiotic-resistance cases. Examples include treatment of antibiotic-resistant *Pseudomonas aeruginosa* ear and wound infections with lytic phages, which have resulted in successful treatment outcomes [169,170]. Another recent well-known example includes infusion into the bloodstream of lytic phages targeting *Acinetobacter baumanii*, resulting in an almost immediate positive response and recovery of the patient [171]. Other approaches involve the administration of phage proteins, particularly endolysins, which degrade the bacterial cell wall and seem to be effective in treating bacterial infections [172]. It has been suggested, however, that phage therapy may provide better results in combination with antibiotic treatment. Similar to the use of multiple antibiotics per treatment, the combination of antibiotic and phage treatment might prolong the development of bacterial resistant to phage therapy [173].

It is becoming evident that virome research is needed to understand how changes in viral communities are associated with disease, dysbiosis, and infections. As new bioinformatic tools and enhanced reference databases provide more accurate strain-level resolution, virus- and virome-directed therapeutic approaches have the potential to be more successful. Although virome-directed therapeutic research is still in the relatively early stages of development, it has been suggested that directly or indirectly altering the virome may improve health outcomes in disease phenotypes associated with virome perturbations. The use of prebiotics (e.g., inulin and fructooligosaccharides) and probiotics, for instance, may indirectly target the virome since these may potentially affect bacterial membership and function. Another virome-directed therapeutic approach may include the utilization of recombinant phages. Phages with genetically enhanced lytic functions have been tested and have shown to be effective [174]. This strategy has also been tested in temperate phages carrying genes conferring sensitivity to streptomycin and nalidixic acid, which have been successfully delivered in resistant strains [175]. A similar approach can be utilized with recombinant temperate phages that could integrate into specific regions within the bacterial genome and disrupt genes involved in bacterial replication, cell structure and virulence gene expression.

## 10. Conclusions and Future Directions

Current knowledge of the gut virome structure and function is providing opportunities to identify cause/effect relationships in maintaining health and in association with dysbiosis and various disease phenotypes. As with bacterial members of the gut microbiome, translational studies are needed to elucidate mechanisms of potential disease progression associated with the gut virome. Future studies may target viral lytic/lysogenic cycles or other pathways to restore gut microbiome balance. Other approaches may include probiotic supplementation with phage-resistant bacteria. This would be particularly beneficial for conditions where bacterial diversity is decreased in association with an increased phage diversity and/or richness. Finally, phages can be (re)used to treat antibiotic-resistant bacteria that cause serious and life-threatening conditions. It is not surprising that a number of future studies will continue to find further associations between the gut virome and other diseases. As more is understood about the gut virome ecology and biology and trans-kingdom interactions, translational and clinical studies can provide evidence of potential therapeutic approaches that may target or utilize these less-studied members of the microbiome to improve health and well-being.

## Figures and Tables

**Figure 1 viruses-11-00656-f001:**
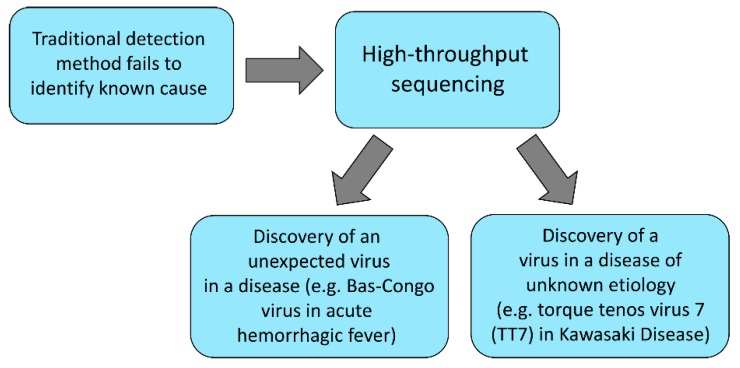
Flow gram of potential applications of high-throughput sequencing in the discovery of an unexpected eukaryotic virus in a disease, and the discovery of eukaryotic viruses in a disease of unknown etiology.

**Figure 2 viruses-11-00656-f002:**
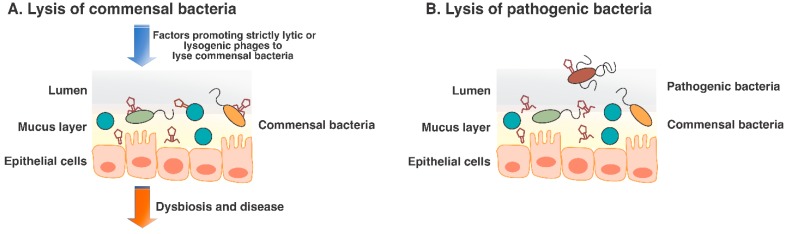
Potential phage-mediated lysis of commensal and pathogenic bacteria in the human gut. Panel (**A**) shows lysis of commensal bacteria in the human gut triggered by external factors that would need to individually be evaluated for each disease phenotype (e.g., Type 1 diabetes (T1D), Inflammatory Bowel Disease (IBD) and cancer). Panel (**B**) shows how phages can potentially aid in pathogen lysis. This is hypothesized based on T4 phages in vitro experiments [36].

**Figure 3 viruses-11-00656-f003:**
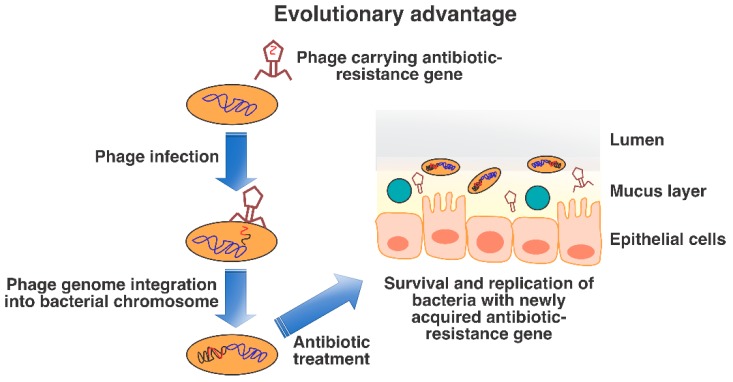
Lysogenic conversion of commensal bacteria. Temperate phages can carry genes that can confer an evolutionary advantage to the bacterial host cell. Figure shows antibiotic-resistance as an example, which can further aid commensal bacteria to survive when exposed to specific antibiotics.

**Figure 4 viruses-11-00656-f004:**
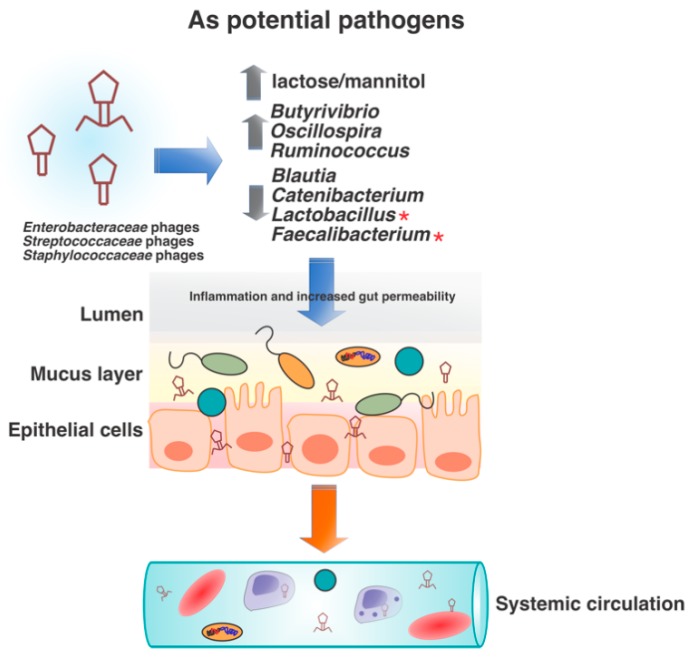
Phages as potential human pathogens. This was probably first suggested with phages harboring stx, which represents a serious risk to human health. Stx B subunit targets a human endothelial cells receptor known as Gb3. Another study involving introducing phages infecting bacteria from the *Enterobacteriaceae*, *Staphylococcaceae* and *Streptococcaceae* families to mice also demonstrated the possibility of phages as human pathogens. An increase in lactulose/mannitol ratio and in *Butyrivibrio*, *Oscillospira* and *Ruminococcus* relative abundances, as well as a decrease in *Blautia*, *Catenibacterium*, *Lactobacillus* and *Faecalibacterium* relative abundances was observed. The decrease in *Lactobacillus* and *Faecalibacterium* (highlighted in red asterisks), specifically, is associated with impaired gut permeability and inflammation.

**Table 1 viruses-11-00656-t001:** Examples of evolutionary advantages conferred through lysogenic conversion. Modified from [60].

Evolutionary Advantage through Lysogenic Conversion	Bacterial Host	Reference
Cell colonization and adhesion	*Escherichia coli*	[69]
	*Pseudomonas aeruginosa*	[70]
	*Streptococcus mitis*	[71]
	*Vibrio cholerae*	[72]
Promotion of cell invasion	*Salmonella enterica*	[73]
	*Streptococcus pyogenes*	[74]
	*Staphylococcus aureus*	[75]
Resistance to serum and phagocytes	*Escherichia coli*	[76]
	*Pseudomonas aeruginosa*	[77]
	*Salmonella enterica*	[78]
	*Shigella dysenteriae*	[79]
	*Staphyloccoccus aureus*	[80]
	*Streptococcus pyogenes*	[81]
Exotoxin production	*Clostridium botulinum*	[82]
	*Corynebacterium diphtheriae*	[83]
	*Escherichia coli*	[84]
	*Pseudomonas aeruginosa*	[85]
	*Shigella dysenteriae*	[86]
	*Staphylococcus aureus*	[87]
	*Streptococcus pyogenes*	[88]
	*Vibrio cholerae*	[89]
Antibiotic susceptibility	*Staphylcoccus aureus*	[90]
	*Streptococcus pyogenes*	[91]

**Table 2 viruses-11-00656-t002:** Bioinformatic tools for virome analyses. Table shows selected bioinformatic tools classified as alignment- or k-mer-based, a brief description and the source.

Bioinformatic Tool	Description	Source	Reference
**Alignment-based**			
VIROME	Web-application interface for the classification of Open-Reading Frames (ORF) or assembled data, which receive one classification.	http://virome.dbi.udel.edu	[146]
VirusSeeker	Linux-based for the classification of sequences at the nucleotide and amino acid level. It is used for virus characterization and discovery; the latter requires assembled reads.	http://pathology.wustl.edu/virusseeker/index.htm	[147]
VirFind	Web-based tool that maps the reads to reference genomes and also performs de novo assembly to get longer contigs to identify known viruses and discover new ones. It performs Blastn and Blastx.	http://virfind.org/j/	[148]
FastViromeExplorer	Pseudo-alignment tool that maps reads to a reference virus database, filters the alignment results based on minimal coverage criteria and reports virus types and abundances along with taxonomic annotation.	https://bench.cs.vt.edu/FastViromeExplorer/	[149]
VirMap	Suitable for low coverage and highly divergent viruses in metagenomic datasets. Uses a mapping assembly algorithm with both nucleotide and amino-acid alignments to build virus-like super-scaffolds. It possesses a taxonomic classification algorithm based on bits-per-base scoring system.	https://github.com/cmmr/VirMAP	[150]
EZ-Map	Python-based to filter, align, and analyze viromes from cell-free DNA samples.	https://github.com/dekoning-lab/ezmap	[151]
MetaVir2	Web-application interface that works with assembled reads to perform taxonomy assignments based on available sequences in RefSeq.	http://metavir-meb.univ-bpclermont.fr	[152]
Vipie	Web-application capable of analyzing datasets from different studies by performing de-novo assembly, followed by taxonomic classification.	https://binf.uta.fi/vipie/	[153]
ViromeScan	Linux-based application that performs taxonomy classification from raw data. The tool uses hierarchical databases for eukaryotic viruses to assign reads to viral species.	http://sourceforge.net/projects/viromescan/	[154]
Vanator	Perl-based pipeline designed for metagenomics and virus discovery projects using Illumina FASTQ-formatted deep sequencing reads as the input.	https://sourceforge.net/projects/vanator-cvr/	[155]
VirSorter	Predicts prophage and viral sequences in a reference-dependent and -independent manner. Detects circular sequences, performs gene prediction, removes poor quality protein-predicted sequences and those remaining are compared to PFAM and RefSeqABVir or Viromes databases.	https://sourceforge.net/projects/viromescan/	[156]
VirusDetect	Performs virus identification by aligning small RNA reads to a known virus reference database and also performs de novo assembly.	http://virusdetect.feilab.net/cgi-bin/virusdetect/index.cgi	[157]
PHASTER	The enhanced release of PHAST for the fast identification and annotation of prophage sequences from assembled metagenomic datasets.	http://phaster.ca	[158]
**K-mer-based**			
MetaVir2	Web-application interface that works with assembled reads to perform taxonomy assignments based on available sequences in RefSeq. Provides users the option to perform the analysis based on k-mers.	http://metavir-meb.univ-bpclermont.fr	[152]
VIP	Developed for identification of viral pathogens from metagenomic datasets. Removes background reads, classifies reads based on nucleotide and amino acid homology, and uses k-mer based de novo assembly for evolutionary studies.	https://github.com/keylabivdc/VIP/blob/master/README.md	[159]
VirFinder	K-mer-based tool to identify sequence signatures that distinguish viral sequences from host sequences.	https://github.com/jessieren/VirFinder	[160]

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
