# Peer review of "Human Virome and Disease: High-Throughput Sequencing for Virus Discovery, Identification of Phage-Bacteria Dysbiosis and Development of Therapeutic Approaches with Emphasis on the Human Gut"

_viruses, 2019, doi:10.3390/v11070656_

Round 1

Reviewer 1 Report

The manuscript by Tasha M. Santiago-Rodriguez and Emily Hollister provides an interesting review of a number of facets of Human virome in disease. In general, the manuscript is well written and will be appreciated by a wide readership. That being said, a number of concepts or terms provide conflicting ideas with the current state of knowledge. I agree with our knowledge about human virome, especially of a eukaryotic virus, is less advanced compared to bacteriome as well as that of bacteriophages. This review mainly focuses on the relationship between gut bacteriome and diseases, then how the virus can affect on disease condition through modifying gut bacteriome, gut dysbiosis (I appreciate the part of Human endogenous retroviruses). However, the part of this review relating eukaryotic viruses is a bit too dismissive and vague and could be improved by modification. The paragraph of “cancer” is placed under “Virome-associated gut dysbiosis”, however, it does not seem right. Viruses such as EBV, HBV, HCV, HIV-1, HPV and HTLV-1 are normally not considered as a component of human virome, because these viruses are more recognised in the relation to a specific disease and/or are not found in a normal healthy population. Yes, “The underlying mechanisms on how these microorganisms promote cancer and the potential therapeutic approaches are a matter of ongoing research.”, however, our knowledge about this is well advanced compared to the relationships between diseases and gut-microbiome, and has provided key ideas on how the virus infection can cause cancers (immunodeficiency, chronic inflammation and virus oncogenes).  Examples of the associations between the gut microbiome and specific bacterial provided in the review are a mixture of concepts, like ‘microbiome and anti-PD-1 therapy’, ‘ chemotherapy and microbiome’ and ‘cervical cancer progress and microbiome (it is even not gut microbiome), and are meaningless providing without interpretation or annotation.

Probably, as long as I believe, HPV could be truly the matter of human virome as a variety of HPV types (high, low risk and asymptomatic/non-pathogenic) are known to be identified in normal healthy population (without infected lesion), and even infected with high-risk HPV, a very small population of infected individuals develops advanced cancer. Furthermore, trans kingdom interaction may be involved in its cancer progression (ref 84).

Please take a look at these papers.

The Human Skin Double-Stranded DNA Virome: Topographical and Temporal Diversity, Genetic Enrichment, and Dynamic Associations with the Host Microbiome

Geoffrey D. Hannigan, Jacquelyn S. Meisel, Amanda S. Tyldsley, Qi Zheng, Brendan P. Hodkinson, Adam J. SanMiguel, Samuel Minot, Frederic D. Bushman, Elizabeth A. Grice

DOI: 10.1128/mBio.01578-15

Composite Analysis of the Virome and Bacteriome of HIV/HPV Co-Infected Women Reveals Proxies for Immunodeficiency

by Juliana D. Siqueira, Gislaine Curty, Deng Xutao, Cristina B. Hofer, Elizabeth S. Machado, Héctor N. Seuánez, Marcelo A. Soares, Eric Delwart and Esmeralda A. Soares

Viruses 2019, 11(5), 422; https://doi.org/10.3390/v11050422

Author Response

Reviewer #1:

The manuscript by Tasha M. Santiago-Rodriguez and Emily Hollister provides an interesting review of a number of facets of Human virome in disease. In general, the manuscript is well written and will be appreciated by a wide readership. That being said, a number of concepts or terms provide conflicting ideas with the current state of knowledge. I agree with our knowledge about human virome, especially of a eukaryotic virus, is less advanced compared to bacteriome as well as that of bacteriophages. This review mainly focuses on the relationship between gut bacteriome and diseases, then how the virus can affect on disease condition through modifying gut bacteriome, gut dysbiosis (I appreciate the part of Human endogenous retroviruses). However, the part of this review relating eukaryotic viruses is a bit too dismissive and vague and could be improved by modification. 

We thank the reviewer and agree that the manuscript can benefit from expanding the section of eukaryotic viruses. We have added an additional paragraph to highlight how high-throughput sequencing can enable the identification and characterization of viral strains that can cause diseases of unknown etiology for example, or those viruses that unexpectedly can cause disease with a different expected etiology (Lines 125-145). We have also included a flow gram of the application of high-throughput sequencing in both of these situations (New Figure 1).

The paragraph of “cancer” is placed under “Virome-associated gut dysbiosis”, however, it does not seem right. Viruses such as EBV, HBV, HCV, HIV-1, HPV and HTLV-1 are normally not considered as a component of human virome, because these viruses are more recognized in the relation to a specific disease and/or are not found in a normal healthy population. Yes, “The underlying mechanisms on how these microorganisms promote cancer and the potential therapeutic approaches are a matter of ongoing research.”, however, our knowledge about this is well advanced compared to the relationships between diseases and gut-microbiome, and has provided key ideas on how the virus infection can cause cancers (immunodeficiency, chronic inflammation and virus oncogenes). Examples of the associations between the gut microbiome and specific bacterial provided in the review are a mixture of concepts, like ‘microbiome and anti-PD-1 therapy’, ‘ chemotherapy and microbiome’ and ‘cervical cancer progress and microbiome (it is even not gut microbiome), and are meaningless providing without interpretation or annotation.

We thank the reviewer for this insight and have now modified the manuscript so that viruses in cancer is a separate version from the version “Virome-associated gut dysbiosis”. We also thank the reviewer for these insights and have included them in the manuscript (lines 437-446). We also understand that a number of the examples provided are not associated with the human gut; thus, we have removed “gut” from the section “Virome-associated gut dysbiosis” and throughout the manuscript so the sections are more inclusive to other human body sites. This was also highlighted (line 61).

Probably, as long as I believe, HPV could be truly the matter of human virome as a variety of HPV types (high, low risk and asymptomatic/non-pathogenic) are known to be identified in normal healthy population (without infected lesion), and even infected with high-risk HPV, a very small population of infected individuals develops advanced cancer. Furthermore, trans kingdom interaction may be involved in its cancer progression (ref 84).

We thank the reviewer for this insight. We have included this in the revised version of the manuscript (lines 437-442).

Please take a look at these papers.

The Human Skin Double-Stranded DNA Virome: Topographical and Temporal Diversity, Genetic Enrichment, and Dynamic Associations with the Host Microbiome Geoffrey D. Hannigan, Jacquelyn S. Meisel, Amanda S. Tyldsley, Qi Zheng, Brendan P. Hodkinson, Adam J. SanMiguel, Samuel Minot, Frederic D. Bushman, Elizabeth A. Grice

DOI: 10.1128/mBio.01578-15

We thank the reviewer for providing this reference. We have included it in the revised version of the manuscript (line 167).

Composite Analysis of the Virome and Bacteriome of HIV/HPV Co-Infected Women Reveals Proxies for Immunodeficiency by Juliana D. Siqueira, Gislaine Curty, Deng Xutao, Cristina B. Hofer, Elizabeth S. Machado, Héctor N. Seuánez, Marcelo A. Soares, Eric Delwart and Esmeralda A. Soares Viruses 2019, 11(5), 422; https://doi.org/10.3390/v11050422

We thank the reviewer for providing this reference. We have included it in the revised version of the manuscript (426-427).

Reviewer 2 Report

This review article is focused on the human “virome”, which includes eukaryotic viruses and bacteriophages that infect bacteria within the human microbiome.  Generally, the article was well-written and interesting.  Nevertheless, the manuscript could be improved.

Major comments:

1.    The article focuses much more heavily on bacteriophages than on eukaryotic viruses that cause human disease.  The title of the article might more accurately reflect the actual content or increase discussion of eukaryotic viruses.

2.    The referencing throughout the text needs to be improved.  The number of references is minimal.

3.    Page 3 – The trans-kingdom interactions are fascinating, particularly the dependence of certain eukaryotic viruses on prokaryotic hosts, but are barely discussed here.

4.    Page 4 or elsewhere - It might be informative to discuss whether lytic or lysogenic phages have a greater effect on the microbiome and human disease.

5.    Page 10 – Line 396 – “The underlying mechanisms on how these microorganisms promote cancer….are a matter of ongoing research”.  This statement is rather dismissive of the entire subject.  All of the viruses mentioned make gene products that contribute directly to cell proliferation, inactivation of tumor suppressor genes, or immune suppression.

Minor comments:

1.    Page 2 – line 74 – What do they mean by HERVs are replicated in a pervasive manner? More accurately, HERVs have not been shown to replicate in any human cell lines, although these viruses clearly replicated and integrated into germ cells in the past.  HERV transcription often is activated during human disease, and their encoded gene products may contribute to disease.

2.    Page 3 – Line 97 -  “Human enteric eukaryotic viruses, particularly, including those from the Phycodnaviridae, Herpesviridae, Poxviridae, Mimiviridae and Iridoviridae families appear to be acquired from an early age”.  No references are provided here.  The cited families all are DNA viruses that do not infect the gut. 

3.    Page 3 – Line 128 – “In further support of diet as a driver of gut virome composition…”.  Is it diet that affects the virome or vice versa?

4.    Page 14 – Line 46 – Should be “antibiotic-resistant bacteria”.

Author Response

Reviewer #2:

This review article is focused on the human “virome”, which includes eukaryotic viruses and bacteriophages that infect bacteria within the human microbiome.  Generally, the article was well-written and interesting. Nevertheless, the manuscript could be improved.

Major comments:

1.     The article focuses much more heavily on bacteriophages than on eukaryotic viruses that cause human disease. The title of the article might more accurately reflect the actual content or increase discussion of eukaryotic viruses.We agree with the reviewer. While we have expanded on the eukaryotic virus fraction (lines 125-145; 478-508), we have modified the title so it could better reflect most of the content of the review. 

2.    The referencing throughout the text needs to be improved. The number of references is minimal.

We thank the reviewer for highlighting this. We have included new references 12-15; 18-19; 22-23; 28; 32-34; 40-41; 43; 44-46; 104, as well as the references from the new section “Trans-kingdom interactions” (lines 478-508; references 119-124), as well as the bioinformatic tools for virome analyses (references 126-140).

3.    Page 3 – The trans-kingdom interactions are fascinating, particularly the dependence of certain eukaryotic viruses on prokaryotic hosts, but are barely discussed here.

We agree with the reviewer in that this section needed to be expanded. We have added an additional paragraph on trans-kingdom interactions (lines 478-508).

4.    Page 4 or elsewhere - It might be informative to discuss whether lytic or lysogenic phages have a greater effect on the microbiome and human disease.

We thank the reviewer for highlighting this. We still do not have substantial information about this subject, probably because of the limitations in annotation tools, which rely mostly on alignments to currently available phage genomes. Most studies so far have shown that most phages across different human sites are lysogenic, mostly due to the identification of various integrases. In some conditions, such as periodontitis, a higher abundance of myoviruses has been observed. Most myoviruses are lytic, and a minority are lysogenic. Although this is limited to available sequences in databases, this may suggest a potential association or role of lytic phages from specific families in disease. It is possible that both lytic and lysogenic phages have an equal impact on certain diseases; however, this would have to be evaluated for each condition individually. We have included this point in the revised version of the manuscript (lines 181-188). 

5.    Page 10 – Line 396 – “The underlying mechanisms on how these microorganisms promote cancer….are a matter of ongoing research”. This statement is rather dismissive of the entire subject.  All of the viruses mentioned make gene products that contribute directly to cell proliferation, inactivation of tumor suppressor genes, or immune suppression.

We agree with the reviewer, have removed the statement and have modified the statements (lines 443-446). 

Minor comments:

1.     Page 2 – line 74 – What do they mean by HERVs are replicated in a pervasive manner? More accurately, HERVs have not been shown to replicate in any human cell lines, although these viruses clearly replicated and integrated into germ cells in the past.  HERV transcription often is activated during human disease, and their encoded gene products may contribute to disease.

We thank the reviewer for highlighting this. We have removed “pervasive manner” to avoid confusion and have modified the statement (lines 82-85).

2.     Page 3 – Line 97 -  “Human enteric eukaryotic viruses, particularly, including those from the Phycodnaviridae, Herpesviridae, Poxviridae, Mimiviridae and Iridoviridae families appear to be acquired from an early age”.  No references are provided here.  The cited families all are DNA viruses that do not infect the gut. 

We thank the reviewer for highlighting this. We have now included the reference for this statement (lines 112-113, new reference 28).

3.     Page 3 – Line 128 – “In further support of diet as a driver of gut virome composition…”.  Is it diet that affects the virome or vice versa?

We thank the reviewer for highlighting this. Since this is relatively hard to determine, we have tone-down the statement (line 162 and 164-165).

4.     Page 14 – Line 46 – Should be “antibiotic-resistant bacteria”

We have modified accordingly from “antibiotic-resistance bacteria” to “antibiotic-resistant bacteria” (line 593).